# Impact of extracorporeal membrane oxygenation-related complications on in-hospital mortality

**Eunae Byun**[1,2], **Pil Je Kang**[3], **Sung Ho Jung**[3], **Seo Young Park**[4], **Sang Ah Lee**[1], **Tae-Won Kwon**[1,5,6], **Yong-Pil Cho**[1] *

1 Division of Vascular Surgery, Department of Surgery, Asan Medical Center, University of Ulsan College of Medicine, Seoul, Republic of Korea, 2 Department of Surgery, Dong-A University Hospital, Busan, Republic of Korea, 3 Department of Thoracic Surgery, Asan Medical Center, University of Ulsan College of Medicine, Seoul, Republic of Korea, 4 Department of Statistics and Data Science, Korea National Open University, Seoul, Republic of Korea, 5 Department of Acute Care Surgery, Korea University Guro Hospital, Seoul, Republic of Korea, 6 Armed Forces Trauma Center, Bundang, Republic of Korea

* ypcho@amc.seoul.kr

## Abstract

### Introduction

Although extracorporeal membrane oxygenation (ECMO) is a well-established treatment for supporting severe cardiopulmonary failure, the morbidity and mortality of patients requiring ECMO support remain high. Evaluating and correcting potential risk factors associated with any ECMO-related complications may improve care and decrease mortality. This study aimed to assess the predictors of ECMO-related vascular and cerebrovascular complications among adult patients and to test the hypothesis that ECMO-related complications are associated with higher in-hospital mortality rates.

### Methods

This single-center, retrospective study included 856 ECMO runs administered via cannulation of the femoral vessels of 769 patients: venoarterial (VA) ECMO ($n = 709$, 82.8%) and venovenous (VV) ECMO ($n = 147$, 17.2%). The study outcomes included the occurrence of ECMO-related vascular and cerebrovascular complications and in-hospital death. The association of ECMO-related complications with the risk of in-hospital death was analyzed.

### Results

The incidences of ECMO-related vascular and cerebrovascular complications were 20.2% and 13.6%, respectively. The overall in-hospital mortality rate was 48.7%: 52.8% among VA ECMO runs and 29.3% among VV ECMO runs. Multivariable analysis indicated that age ($P < 0.01$), cardiopulmonary cerebral resuscitation ($P < 0.01$), continuous renal replacement therapy ($P < 0.01$), and initial platelet count [<50×10$^3$/μL ($P = 0.02$) and 50–100(×10$^3$)/μL ($P < 0.01$)] were associated with an increased risk of in-hospital death. ECMO-related vascular and cerebrovascular complications were not independently associated with higher in-hospital mortality rates for VA or VV ECMO runs.

**Data Availability Statement:** All relevant data are within the manuscript and its Supporting information files.

**Funding:** The author(s) received no specific funding for this work.

**Competing interests:** The authors have declared that no competing interests exist.

## Conclusion

ECMO-related vascular and cerebrovascular complications were not associated with an increased risk of in-hospital death among adult patients.

## Introduction

Recent technical advances have led to increasingly wider adoption of extracorporeal membrane oxygenation (ECMO) for temporary mechanical cardiopulmonary support for patients whose heart and lungs can no longer provide adequate physiologic support [1–3]. ECMO can be either venoarterial (VA) for systemic circulatory support—providing oxygenation and circulatory support—or venovenous (VV) for gas exchange without hemodynamic support— providing ventilation in addition to isolated oxygenation (S1 Fig) [1, 2]. Currently, ECMO is a well-established treatment for temporarily supporting patients with severe cardiopulmonary failure. However, despite increasing experience with ECMO and recent technical improvements, the morbidity and mortality among patients undergoing ECMO support remain high because all ECMO patients are critically ill by the time they require this intervention. Vascular and cerebrovascular complications during ECMO support—with reported rates of 30% to 60% of patients [4, 5]—are common among high-risk patients, including those who are critically ill, exposed to anticoagulation, or susceptible to coagulopathy and platelet (PLT) dysfunction [6]. These complications can arise from various contributing factors [7, 8]. In this study, we aimed to identify predictors of ECMO-related vascular and cerebrovascular complications among adult patients and to test the hypothesis that ECMO-related complications are associated with an increased risk of in-hospital death.

## Materials and methods

### Study design and study sample

In this single-center, retrospective, observational study, we analyzed the data extracted from the prospectively collected ECMO registry of patients ≥ 18 years of age who underwent ECMO at our hospital for support during severe cardiac and/or pulmonary failure. Further clinical details were obtained from retrospective reviews of patients' medical records. Approval for data collection and publication was obtained from the institutional review board of the Asan Medical Center (IRB No. 2020–0793), which waived the requirement for written informed consent because of the study's retrospective design. All methods were performed in accordance with the relevant guidelines and regulations.

Between January 2015 and December 2019, 868 ECMO patients were screened for inclusion in this study. The exclusion criteria were as follows: 1) ECMO implanted at other hospitals ($n$ = 50), 2) only central ECMO implantation without peripheral cannulation ($n$ = 40), and 3) ECMO running time less than 20 min ($n$ = 9) [2, 7, 8]. Our analysis was not performed on a per-patient basis but on a per-ECMO basis. Among the included patients, nine underwent both VA and VV ECMO runs and were included in both the VA and VV ECMO groups. A total of 856 ECMO runs—administered via cannulation of the femoral vessels—from 769 patients were included in the final analysis. Eligible ECMO procedures were stratified into 2 groups according to the mode of ECMO: VA ECMO ($n$ = 709, 82.8%) and VV ECMO ($n$ = 147, 17.2%) (S2 Fig).

For all consecutive patients, we entered the following data into an Excel (Microsoft Corp., Redmond, WA, USA) database for retrospective analysis: demographics, underlying diseases, risk factors of interest (including a history of cardiovascular disease and interventions), clinical characteristics, laboratory profiles, indications for ECMO support, ECMO-related complications,

in-hospital deaths, and causes of death. We also collected and analyzed ECMO data from the ECMO records: running time, cannula size, and placement of distal perfusion catheter.

## ECMO protocol

The ECMO protocol is detailed in the S1 Text.

## Definitions and study outcomes

The study outcomes included the occurrence of ECMO-related vascular complications (defined as thromboembolic, bleeding, deep venous thrombosis [DVT], and cannula-related local vascular complications), ECMO-related cerebrovascular complications, and in-hospital death. Thromboembolic complications—both in the patient or the circuit—were defined as a medical diagnosis of limb ischemia, intracardiac or aortic thrombosis, or visible thrombosis of the oxygenator and circuits confirmed by ultrasonography or computed tomographic (CT) scan [3]. Bleeding complications were defined as loss of more than 2.0 g/dL hemoglobin (Hb) in 24 h or a bleeding rate of more than 20 mL/kg per day, blood transfusion of more than 10 mL/kg per day, or any bleeding requiring radiologic intervention or surgical treatment [3]. DVT was defined as a medical diagnosis of venous thrombosis using ultrasonography or CT scan, or pulmonary embolism using CT angiography of the pulmonary arteries or a ventila-tion-perfusion lung scan. Cannula-related local vascular complications included pseudoaneur-ysm, arteriovenous fistula, and arterial dissection. All ECMO-related vascular complications were managed by experienced specialized vascular surgeons. Cerebrovascular complications were defined as either embolic or hemorrhagic strokes reported on CT scans with no other potential cause, confirmed by an expert neurologist [6].

Data for all outcomes were centrally reviewed and blindly adjudicated by two experienced independent physicians. For quality control, second reviews were conducted by other physi-cians blinded to the original adjudicated results. Every event of each outcome was analyzed individually. Then, the association of ECMO-related vascular and cerebrovascular complica-tions with the risk of in-hospital death was analyzed.

## Statistical analysis

Continuous variables are reported as mean and standard deviation or median and interquartile range based on their distribution. Categorical variables are presented as count and proportion (%). All of the outcomes are binary variables; thus, multivariable logistic regression was used to investigate the explanatory variables and outcomes. To account for the clustering effect within each patient, we used the generalized estimation equation method to make inferences about the regression coefficients. To select the variables to be included in the multivariable model, we considered the statistical significance ($P < 0.1$) from the univariable analysis, as well as clinical knowledge, multicollinearity among the variables, and the parsimony of the model to avoid overfitting. $P$-values $\leq 0.05$ were considered statistically significant for individual tests. All analyses were conducted using R 4.0.4 (R Foundation for Statistical Computing, Vienna, Austria. https://www.R-project.org/).

## Results

### Clinical characteristics and outcomes

Among the study sample of 856 ECMO runs, the VA ECMO group consisted of 709 ECMO runs (82.8%) and the VV ECMO group consisted of 147 ECMO runs (17.2%). The baseline and clinical characteristics of the study sample are presented in S1 Table. Four-hundred and

**Table 1. ECMO-related vascular, cerebrovascular, and other complications, stratified by ECMO mode.**

|  | Total | VA ECMO | VV ECMO | *P*-value |
|---|---|---|---|---|
| **Number of ECMO runs** | 856 | 709 (82.8) | 147 (17.2) |  |
| **Any complications** | 297 (34.7) | 259 (36.5) | 38 (25.9) | 0.02 |
| **ECMO-related vascular**[a] | 173 (20.2) | 154 (21.7) | 19 (12.9) | 0.02 |
| Thromboembolic | 54 (6.3) | 52 (7.3)[b] | 2 (1.4) | 0.01 |
| Bleeding | 77 (9.0) | 65 (9.2) | 12 (8.2) | 0.82 |
| Deep venous thrombosis | 17 (2.0) | 14 (2.0) | 3 (2.0) | >0.99 |
| Cannula-related local | 24 (2.8) | 22 (3.1) | 2 (1.4) | 0.37 |
| Pseudoaneurysm | 14 (1.6) | 12 (1.7) | 2 (1.4) | >0.99 |
| Arteriovenous fistula | 9 (1.1) | 9 (1.3) | 0 (0.0) | 0.35 |
| Arterial dissection | 1 (0.1) | 1 (0.1) | 0 (0.0) | >0.99 |
| **ECMO-related cerebrovascular** | 116 (13.6) | 103 (14.5) | 13 (8.8) | 0.09 |
| **Others** |  |  |  |  |
| Local wound infection | 52 (6.1) | 46 (6.5) | 6 (4.1) | 0.36 |
| Peripheral neurologic | 28 (3.3) | 23 (3.2) | 5 (3.4) | >0.99 |
| Bowel ischemia | 16 (1.9) | 15 (2.1) | 1 (0.7) | 0.40 |
| Pneumothorax | 1 (0.1) | 0 (0.0) | 1 (0.7) | 0.38 |

Values in parentheses are percentages.

ECMO, extracorporeal membrane oxygenation; VA, venoarterial; VV, venovenous.

[a] Any event of each ECMO-related vascular complication was included individually.

[b] Included six major amputations as a result of limb ischemia.

sixty-one of 856 ECMO runs (53.9%) were successfully weaned. The weaning rate in the VV ECMO group was significantly higher than that in the VA ECMO group (51.6% vs. 64.6%, $P < 0.01$): among the successfully weaned-off ECMO runs, 10.4% (38/366), 7.7% (28/366), and 3.6% (13/366) were bridged to heart transplantation, open heart surgery, and other forms of cardiac support, respectively, in the VA ECMO group, and 23.2% (22/95) and 3.2% (3/95) were bridged to lung transplantation and other forms of respiratory support, respectively, in the VV ECMO group. The in-hospital mortality rate was 48.7% (52.8% in the VA ECMO group and 29.3% in the VV ECMO group, $P < 0.01$). Although patients in the VA ECMO group were more likely to have medical comorbidities than those in the VV ECMO group, the mean duration of hospitalization and intensive care unit admission were comparatively longer in the VV ECMO group.

The incidences of ECMO-related vascular and cerebrovascular complications were 20.2% and 13.6%, respectively. The rate of ECMO-related vascular complications in the VA ECMO group was significantly higher than that in the VV ECMO group (21.7% vs. 12.9%, $P = 0.02$), whereas there was no significant difference in the incidence of ECMO-related cerebrovascular complications between the two groups (14.5% vs. 8.8%, $P = 0.09$) (Table 1). For the analyses of the individual components of the ECMO-related vascular complications, the rate of thrombo-embolic complications in the VA ECMO group was significantly higher than that in the VV ECMO group (7.3% vs. 1.4%, $P = 0.01$), and there was no significant difference in the bleeding (9.2% vs. 8.2%, $P = 0.82$) and DVT (2.0% vs. 2.0%, $P > 0.99$) rates between the two groups. Limb ischemia ($n = 54$) was the only thromboembolic complication, and in the VA ECMO group, we had to perform six major amputations (6/709, 0.8%) because of limb ischemia. An antegrade distal perfusion catheter to augment extremity perfusion was selectively placed in 31.6% of the VA ECMO runs (224/709) for patients with peripheral arterial occlusive disease

(PAOD) or in whom we suspected limb ischemia after ECMO implantation; 10.3% of these patients (23/224) eventually developed limb ischemia, among whom we performed five major amputations. Despite the higher rates of in-hospital death and ECMO-related complications in the VA ECMO group, there was no difference in the rate of in-hospital death caused by ECMO-related complications between the groups (6.1% vs. 2.3%, $P$ = 0.49) (S1 Table). The in-hospital mortality rates are presented in S2 Table by the indications for ECMO support.

## Variables associated with ECMO-related vascular and cerebrovascular complications

Multivariable analysis adjusting for confounding variables indicated that PAOD (odds ratio [OR], 3.28; 95% confidence interval [CI], 1.02–10.58; $P$ = 0.047) and ECMO running time (OR, 1.02; 95% CI, 1.01–1.03; $P$ < 0.01) were negative independent risk factors for ECMO-related vascular complications associated with VA ECMO (Table 2). In terms of the incidence of the individual components of ECMO-related vascular complications associated with VA ECMO, age (OR, 0.98; 95% CI, 0.95–1.00; $P$ = 0.02) and continuous renal replacement therapy (CRRT) (OR, 5.27; 95% CI, 1.66–16.77; $P$ < 0.01) were associated with an increased risk of thromboembolic complications (S3 Table), whereas ECMO running time (OR, 1.01; 95% CI, 1.00–1.02; $P$ < 0.01), arterial cannula size (OR, 0.79; 95% CI, 0.63–0.99; $P$ = 0.04), initial Hb level < 8.0 g/dL (OR, 4.19; 95% CI, 2.21–7.96; $P$ < 0.01), and PLT count (50–100[$\times 10^3$]/μL) (OR, 2.07; 95% CI, 1.13–3.79; $P$ = 0.02) were associated with bleeding complications (S4 Table). For VV ECMO, the univariable analysis found that female sex (OR, 2.99; 95% CI, 1.01–

**Table 2. Factors associated with ECMO-related vascular complications in VA ECMO.**

| | Univariable analysis OR (95% CI) | *P*-value | Multivariable analysis OR (95% CI) | *P*-value |
|---|---|---|---|---|
| **Age** | 0.99 (0.98–0.99) | 0.03 | 0.99 (0.98–1.01) | 0.34 |
| **Female** | 1.09 (0.73–1.63) | 0.68 | NA | NA |
| **BMI** | 1.03 (0.97–1.09) | 0.32 | NA | NA |
| **Hypertension** | 0.72 (0.49–1.05) | 0.09 | 0.78 (0.48–1.26) | 0.31 |
| **Diabetes mellitus** | 0.76 (0.50–1.17) | 0.21 | NA | NA |
| **Smoking** | 1.13 (0.73–1.74) | 0.58 | NA | NA |
| **PAOD** | 2.80 (1.05–7.51) | 0.04 | 3.28(1.02–10.58) | 0.047 |
| **History of CAD** | 0.95 (0.61–1.47) | 0.81 | NA | NA |
| **History of CVA** | 1.08 (0.55–2.11) | 0.82 | NA | NA |
| **History of CKD** | 1.33 (0.81–2.20) | 0.26 | NA | NA |
| **CPCR** | 1.45 (0.88–2.39) | 0.15 | NA | NA |
| **CRRT** | 1.70 (1.10–2.62) | 0.02 | 1.48 (0.91–2.38) | 0.11 |
| **ECMO running time (10 h)** | 1.02 (1.01–1.03) | <0.01 | 1.02 (1.01–1.03) | <0.01 |
| **Arterial cannula size** | 0.96 (0.83–1.12) | 0.63 | NA | NA |
| **Initial Hb(ref. $\geq$ 10.0 g/dL)** | | 0.04 | | 0.21 |
| <8.0 g/dL | 2.94 (1.27–6.80) | 0.01 | 2.50 (0.90–6.97) | 0.08 |
| 8.0–10.0 g/dL | 1.17 (0.67–2.04) | 0.58 | 1.03 (0.54–1.95) | 0.93 |
| **Initial PLT(ref. $\geq$ 100$\times 10^3$/μL)** | | 0.16 | | |
| <50$\times 10^3$/μL | 1.69 (0.56–5.10) | 0.36 | NA | NA |
| 50–100($\times 10^3$)/μL | 1.41 (0.94–2.10) | 0.10 | NA | NA |

BMI, body mass index; CPCR, cardiopulmonary cerebral resuscitation; CVA, cerebrovascular accident; CKD, chronic kidney disease; CI, confidence interval; CRRT, continuous renal replacement therapy; CAD, coronary artery disease; ECMO, extracorporeal membrane oxygenation; Hb, hemoglobin; NA, not applicable; OR, odds ratio; PAOD, peripheral arterial occlusive disease; PLT, platelet; ref., reference range; VA, venoarterial.

8.86; *P* = 0.048), a history of chronic kidney disease (CKD) (OR, 0.00; 95% CI, 0.00–0.00; *P* < 0.01), ECMO running time (OR, 1.02; 95% CI, 1.01–1.04; *P* < 0.01), and initial Hb level < 8.0 g/dL (OR, 4.93; 95% CI, 1.21–20.07; *P* = 0.03) were associated with an increased risk of ECMO-related vascular complications (S5 Table). The low number of events (*n* = 19) precluded the application of multivariable analysis.

The univariable analysis for the incidence of ECMO-related cerebrovascular complications associated with VA ECMO identified no significant risk factors. For the variables associated with ECMO-related cerebrovascular complications in VV ECMO, the univariable analysis indicated diabetes mellitus (OR, 3.80; 95% CI, 1.10–13.15; *P* = 0.04), PAOD (OR, 0.00; 95% CI, 0.00–0.00; *P* < 0.01), a history of CKD (OR, 7.59; 95% CI, 1.87–30.75; *P* < 0.01), and cardio-pulmonary cerebral resuscitation (CPCR) (OR, 0.00; 95% CI, 0.00–0.00; *P* < 0.01) were independent predictors of an increased risk of cerebrovascular complications (S6 Table): multivariable models could not be constructed due to the low number of events (*n* = 13).

### Association between ECMO-related complications and the risk of in-hospital death

Among a total of 856 ECMO runs from 769 patients, the overall in-hospital mortality rate was 48.7% (417/856): 52.8% among the VA ECMO runs (374/709) and 29.3% among the VV ECMO runs (43/147). For both VA and VV ECMO runs, multivariable analysis indicated that age (OR, 1.03; 95% CI, 1.01–1.04; *P* < 0.01), CPCR (OR, 1.42; 95% CI, 1.12–1.79; *P* < 0.01), CRRT (OR, 3.57; 95% CI, 2.59–4.93; *P* < 0.01), initial PLT count < 50×10³/μL (OR, 1.60; 95% CI, 1.10–2.33; *P* = 0.02), and initial PLT count 50–100(×10³)/μL (OR, 1.24; 95% CI, 1.07–1.44; *P* < 0.01) were associated with higher in-hospital mortality rates (Table 3). Among the patients in the VA ECMO group, multivariable analysis indicated that age (OR, 1.02; 95% CI, 1.01–1.04; *P* < 0.01), CRRT (OR, 2.93; 95% CI, 2.06–4.17; *P* < 0.01), and initial PLT count (50–100 [×10³]/μL) (OR, 1.64; 95% CI, 1.06–2.54; *P* = 0.03) were negative independent risk factors for in-hospital death (Table 4), whereas among the patients in the VV ECMO group, age (OR, 1.03; 95% CI, 1.00–1.06; *P* = 0.03) and CRRT (OR, 5.48; 95% CI, 2.46–12.23; *P* < 0.01) were associated with an increased risk of in-hospital death (Table 5). In these analyses for all ECMO runs, VA ECMO and VV ECMO, ECMO-related vascular and cerebrovascular complications were not associated with higher in-hospital mortality rates.

### Discussion

Although ECMO is a complex and high-risk therapy [9, 10], this modality has been increasingly adopted after the landmark CESAR (Conventional Ventilatory Support versus Extracorporeal Membrane Oxygenation for Severe Adult Respiratory Failure) and EOLIA (ECMO to Rescue Lung Injury in Severe ARDS) trials, in which ECMO appeared to be superior to ventilator use in the adult population [10, 11]. Whereas VV ECMO replaces failing lungs, VA ECMO replaces both the heart and lungs. To date, few randomized controlled trials have investigated ECMO administration for critically ill patients, and the evidence supporting current ECMO guidelines is weak [12]; however, the indications for ECMO support have continued to evolve over the past decade [2]. Regarding VA ECMO, the primary indication has shifted from post-cardiotomy shock to multifactorial cardiogenic shock and/or cardiac arrest [2]. The proportion of post-cardiotomy shock patients supported by VA ECMO decreased from 56.9% in 2002 to 37.9% in 2012. During this same period, the number of adult patients with cardiopulmonary failure supported by VA ECMO substantially increased [13]. VV ECMO is mainly used to support patients with severe acute respiratory distress syndrome [14] or as a bridging strategy to lung transplantation [15].

**Table 3. Factors associated with in-hospital mortality in both VA and VV ECMO.**

| | Univariable analysis OR(95% CI) | P-value | Multivariable analysis OR (95% CI) | P-value |
|---|---|---|---|---|
| **Age** | 1.02 (1.01–1.03) | <0.01 | 1.03 (1.01–1.04) | <0.01 |
| **Female** | 1.23 (0.91–1.67) | 0.18 | NA | NA |
| **BMI** | 1.01 (0.99–1.03) | 0.65 | NA | NA |
| **Hypertension** | 1.56 (1.17–2.09) | <0.01 | 1.16 (0.82–1.64) | 0.41 |
| **Diabetes mellitus** | 1.41 (1.03–1.94) | 0.03 | 0.97 (0.65–1.46) | 0.89 |
| **Smoking** | 0.91 (0.65–1.28) | 0.60 | NA | NA |
| **PAOD** | 0.60 (0.24–1.51) | 0.28 | NA | NA |
| **History of CAD** | 1.21 (0.85–1.71) | 0.29 | NA | NA |
| **History of CVA** | 1.62 (0.92–2.84) | 0.095 | 1.18 (0.60–2.35) | 0.63 |
| **History of CKD** | 1.52 (1.00–2.31) | 0.051 | NA | NA |
| **CPCR** | 1.19 (1.08–1.32) | <0.01 | 1.42 (1.12–1.79) | <0.01 |
| **CRRT** | 3.93 (2.89–5.35) | <0.01 | 3.57 (2.59–4.93) | <0.01 |
| **ECMO running time (10 h)** | 1.00 (1.00–1.00) | 0.051 | 1.00 (1.00–1.01) | 0.07 |
| **Initial Hb (ref. ≥10.0 g/dL)** | | <0.01 | | 0.03 |
| <8.0 g/dL | 1.24 (1.08–1.42) | <0.01 | 1.06 (0.91–1.24) | 0.47 |
| 8.0–10.0 g/dL | 1.18 (1.08–1.30) | <0.01 | 1.08 (0.96–1.22) | 0.19 |
| **Initial PLT (ref. ≥ 100×10³/μL)** | | <0.01 | | 0.12 |
| <50×10³/μL | 1.37 (1.14–1.64) | <0.01 | 1.60 (1.10–2.33) | 0.02 |
| 50–100(×10³)/μL | 1.18 (1.09–1.27) | <0.01 | 1.24 (1.07–1.44) | <0.01 |
| **ECMO-related vascular complications** | 1.00 (0.85–1.17) | 0.95 | | |
| Thromboembolic | 1.25 (0.99–1.58) | 0.06 | 1.09 (0.75–1.59) | 0.66 |
| Bleeding | 1.21 (0.98–1.50) | 0.08 | 1.03 (0.71–1.48) | 0.89 |
| **Cerebrovascular complications** | 1.27 (0.97–1.67) | 0.08 | 1.16 (0.57–2.32) | 0.69 |

BMI, body mass index; CPCR, cardiopulmonary cerebral resuscitation; CVA, cerebrovascular accident; CKD, chronic kidney disease; CI, confidence interval; CRRT, continuous renal replacement therapy; CAD, coronary artery disease; ECMO, extracorporeal membrane oxygenation; Hb, hemoglobin; NA, not applicable; OR, odds ratio; PAOD, peripheral arterial occlusive disease; PLT, platelet; ref., reference range; VA, venoarterial.

Despite increasing experience with ECMO and recent technical advances, the morbidity and mortality of patients receiving ECMO remain high, with variations between centers, patient subgroups, and indications [6, 16]. ECMO outcomes are influenced not only by factors independent of ECMO (severity and type of patient illness, as well as other organ support) but also by potential ECMO-related complications [6]. Evaluations and clarification of the impact of correctable ECMO-related complications on outcomes could inform safer care and improve outcomes. However, because of the lack of randomized trials, observational series may provide some useful information on factors associated with morbidity and mortality. ECMO-related complications may be mechanical (relating to the ECMO circuit components) or medical [17]. Recent technical improvements—the introduction of centrifugal pumps, low-resistance poly-methylpentene membranes, and modern heparin-coated surfaces, for example—could reduce mechanical complications and hemolysis, but medical complications frequently occur, including vascular and neurological complications [6]. In our analysis, for both modes of ECMO, ECMO-related vascular complications were the most common, followed by cerebrovascular complications.

**Table 4. Factors associated with in-hospital mortality in VA ECMO.**

| | Univariable analysis OR(95% CI) | *P*-value | Multivariable analysis OR (95% CI) | *P*-value |
|---|---|---|---|---|
| **Age** | 1.02 (1.01–1.03) | <0.01 | 1.02 (1.01–1.04) | <0.01 |
| **Female** | 1.34 (0.95–1.90) | 0.095 | 1.48 (0.98–2.22) | 0.06 |
| **BMI** | 1.01 (0.98–1.03) | 0.71 | NA | NA |
| **Hypertension** | 1.50 (1.09–2.06) | 0.01 | NA | NA |
| **Diabetes mellitus** | 1.26 (0.89–1.79) | 0.19 | NA | NA |
| **Smoking** | 0.92 (0.64–1.33) | 0.67 | NA | NA |
| **PAOD** | 0.60 (0.23–1.58) | 0.30 | NA | NA |
| **History of CAD** | 1.01 (0.70–1.46) | 0.95 | NA | NA |
| **History of CVA** | 1.57 (0.86–2.85) | 0.14 | NA | NA |
| **History of CKD** | 1.42 (0.90–2.23) | 0.13 | NA | NA |
| **CPCR** | 1.22 (1.02–1.46) | 0.03 | 1.53 (0.47–4.97) | 0.48 |
| **CRRT** | 3.44 (2.44–4.84) | <0.01 | 2.93 (2.06–4.17) | <0.01 |
| **ECMO running time(10 h)** | 1.00 (1.00–1.00) | 0.48 | NA | NA |
| **Arterial cannula size** | 1.00 (0.94–1.07) | 0.90 | NA | NA |
| **Initial Hb(ref. $\geq$ 10.0 g/dL)** | | <0.01 | | 0.06 |
| <8.0 g/dL | 1.32 (1.05–1.65) | 0.02 | 1.01 (0.52–1.97) | 0.97 |
| 8.0–10.0 g/dL | 1.26 (1.10–1.45) | <0.01 | 1.05 (0.75–1.46) | 0.78 |
| **Initial PLT (ref. $\geq$ 100×10$^3$/μL)** | | <0.01 | | 0.06 |
| <50×10$^3$/μL | 1.66 (1.23–2.22) | <0.01 | 1.92 (0.47–7.86) | 0.36 |
| 50–100(×10$^3$)/μL | 1.36 (1.19–1.56) | <0.01 | 1.64 (1.06–2.54) | 0.03 |
| **ECMO-related vascular complications** | 0.85 (0.67–1.08) | 0.19 | NA | NA |
| Thromboembolic | 1.30 (0.94–1.80) | 0.11 | NA | NA |
| Bleeding | 1.18 (0.83–1.67) | 0.35 | NA | NA |
| **Cerebrovascular complications** | 1.00 (0.94–1.07) | 0.25 | NA | NA |

BMI, body mass index; CPCR, cardiopulmonary cerebral resuscitation; CVA, cerebrovascular accident; CKD, chronic kidney disease; CI, confidence interval; CRRT, continuous renal replacement therapy; CAD, coronary artery disease; ECMO, extracorporeal membrane oxygenation; Hb, hemoglobin; NA, not applicable; OR, odds ratio; PAOD, peripheral arterial occlusive disease; PLT, platelet; ref., reference range; VA, venoarterial.

For ECMO support, cannulation under systemic anticoagulation with a large cannula is critically important and known to be associated with vascular complications, particularly among small women or in association with high systemic vascular resistance [2]. It is generally stated that unless the vessel is at least 1 to 2 mm larger than the cannula in arterial cannulation for VA ECMO, there is an increased risk of limb ischemia; however, the size of the cannula required for ECMO cannulation has not been clearly defined in currently available publications, and it is usual practice to select a cannula that will yield optimal support for a given patient [2]. Currently, the use of prophylactic antegrade perfusion catheters is recommended to reduce the incidence of limb ischemia [18, 19]. Lamb et al. [18] found that 0 of 55 patients with distal perfusion catheters placed prophylactically developed limb ischemia, and in a meta-analysis by Juo et al. [19], distal perfusion catheter placement was associated with a relative risk ratio of limb ischemia of 0.41. In our study, we selectively used a distal perfusion catheter in 31.6% of VA ECMO runs (224/709) for high-risk patients with limb ischemia, and 10.3% of them (23/224) eventually developed limb ischemia. Among the 485 ECMO runs that did not use distal perfusion catheters, there was a 6.0% limb ischemia rate (29/485). Additional studies are required to better understand the role of an antegrade catheter for augmenting extremity perfusion in the prevention of limb ischemia.

**Table 5. Factors associated with in-hospital mortality in VV ECMO.**

| | Univariable analysis OR(95% CI) | *P*-value | Multivariable analysis OR (95% CI) | *P*-value |
|---|---|---|---|---|
| **Age** | 1.02 (0.99–1.05) | 0.18 | 1.03 (1.00–1.06) | 0.03 |
| **Female** | 0.93 (0.44–1.94) | 0.84 | NA | NA |
| **BMI** | 0.99 (0.90–1.08) | 0.80 | NA | NA |
| **Hypertension** | 1.35 (0.65–2.82) | 0.42 | NA | NA |
| **Diabetes mellitus** | 1.63 (0.68–3.92) | 0.27 | NA | NA |
| **Smoking** | 0.83 (0.35–1.99) | 0.68 | NA | NA |
| **PAOD** | Not calculable | | | |
| **History of CAD** | 0.80 (0.15–4.29) | 0.80 | NA | NA |
| **History of CVA** | Not calculable | | | |
| **History of CKD** | 1.50 (0.45–5.01) | 0.51 | NA | NA |
| **CPCR** | 1.06 (0.34–3.26) | 0.92 | NA | NA |
| **CRRT** | 4.77 (2.24–10.17) | <0.01 | 5.48 (2.46–12.23) | <0.01 |
| **ECMO running time(10 h)** | Not calculable | | | |
| **Initial Hb (ref. ≥ 10.0 g/dL)** | Not calculable | | | |
| <8.0 g/dL | | | | |
| 8.0–10.0 g/dL | | | | |
| **Initial PLT (ref. ≥ 100×10³/μL)** | | 0.40 | | |
| <50×10³/μL | 8.37 (0.38–183.14) | 0.18 | NA | NA |
| 50–100(×10³)/μL | 1.34 (0.37–4.93) | 0.66 | NA | NA |
| **ECMO-related vascular complications** | 2.61 (0.65–10.44) | 0.18 | 2.08 (0.44–9.75) | 0.35 |
| Thromboembolic | Not calculable | | | |
| Bleeding | Not calculable | | | |
| **Cerebrovascular complications** | 2.18 (0.66–7.19) | 0.20 | NA | NA |

BMI, body mass index; CPCR, cardiopulmonary cerebral resuscitation; CVA, cerebrovascular accident; CKD, chronic kidney disease; CI, confidence interval; CRRT, continuous renal replacement therapy; CAD, coronary artery disease; ECMO, extracorporeal membrane oxygenation; Hb, hemoglobin; NA, not applicable; OR, odds ratio; PAOD, peripheral arterial occlusive disease; PLT, platelet; ref., reference range; VV, venovenous.

The causes of cerebrovascular complications—ischemic and hemorrhagic strokes—are multifactorial, with thromboembolic events, systemic anticoagulation, and hemodynamic instability thought to contribute [20]. The presence of the circuit adds risk secondary to particles, bubbles, or emboli, which may be inadvertently infused into the arterial circuit. Furthermore, thrombi can form spontaneously in the left atrium and left ventricle due to the low-flow state. The incidence of neurological complications reported in the literature is highly variable (between 4% and 37%) [6, 20]. It may vary according to ECMO indications, types of cannulation, and patient comorbidities.

Given the different indications for ECMO support and the diversity of each patient's underlying disease, deaths among ECMO patients are usually multifactorial; potential predictors of death include older age, female sex, longer support time, decreased cardiac function at baseline [21], high lactate concentration, peripheral vascular disease, chronic obstructive lung disease, renal dysfunction [22, 23], stroke, infection, hypoglycemia, alkalosis [24], device insertion during CPCR, and decreased urine output [25]. In our study, we found that ECMO-related vascular and cerebrovascular complications are common: 20.2% and 13.6% of ECMO runs were associated with ECMO-related vascular and cerebrovascular complications, respectively. Older age, CPCR, CRRT, and initial PLT count were associated with higher in-hospital mortality rates. Similar to a previous study by Bisdas et al. [26], which found that vascular

complications were not associated with mortality in a cohort of 174 VV and VA ECMO runs, our study indicated that ECMO-related vascular and cerebrovascular complications were not associated with an increased risk of in-hospital death associated with either VA or VV ECMO. According to the indications for ECMO support, 87.5% (7/8) of the patients placed on VA ECMO for septic shock did not survive. Although we could not perform statistical analysis due to the small number of patients with septic shock, patients placed on VA ECMO for septic shock consistently have higher mortality rates than other indications.

Our study had important limitations that should be acknowledged. First, although the ECMO data and patient characteristics were collected prospectively, this was a retrospective study subject to selection bias, and some clinical information was not available from the medical records: we could not assess long-term functional status in patients with cerebrovascular complications. Second, this was a single-center study with a relatively small sample size, precluding detailed subgroup analyses, and thus, it is likely to be underpowered for some outcomes. Furthermore, because of various indications for ECMO support, we could not analyze the relationship between indications for ECMO support and outcomes. Third, in some cases, we used alternative definitions from those used by previous publications. The lack of standard definitions means that there are varying indications for ECMO support and definitions of ECMO-related complications among studies; therefore, we cannot directly compare our results with previously published data. Fourth, given the small sample size and the retrospective study design, this study was likely underpowered to provide robust evidence. Future multicenter studies with larger sample sizes and strict, well-defined indications are required to evaluate the impact of ECMO-related complications on in-hospital mortality.

In conclusion, given the heterogeneity—varying indications and underlying diseases—of the study sample, various factors were associated with a higher risk of ECMO-related vascular and cerebrovascular complications. Older age and CRRT were important risk factors for in-hospital death for both VA and VV ECMO runs. However, ECMO-related vascular and cerebrovascular complications were not associated with an increased risk of in-hospital death.

## Supporting information

**S1 Fig. Schematic representative figures of VA and VV ECMO.**
(PDF)

**S2 Fig. Flow chart of ECMO inclusion.**
(PDF)

**S1 Text. Extracorporeal membrane oxygenation (ECMO) protocol.**
(PDF)

**S1 Table. Baseline and clinical characteristics, stratified by the ECMO mode.**
(PDF)

**S2 Table. In-hospital mortality rates according to ECMO indications.**
(PDF)

**S3 Table. Factors associated with ECMO-related thromboembolic complications in VA ECMO.**
(PDF)

**S4 Table. Factors associated with ECMO-related bleeding complications in VA ECMO.**
(PDF)

**S5 Table. Factors associated with ECMO-related vascular complications in VV ECMO.**
(PDF)

**S6 Table. Factors associated with ECMO-related cerebrovascular complications in VV ECMO.**
(PDF)

**S1 Data.**
(XLSX)

## Author Contributions

**Conceptualization:** Eunae Byun, Seo Young Park, Yong-Pil Cho.

**Data curation:** Eunae Byun, Pil Je Kang, Sung Ho Jung, Sang Ah Lee, Tae-Won Kwon, Yong-Pil Cho.

**Formal analysis:** Eunae Byun, Pil Je Kang, Sung Ho Jung, Seo Young Park, Sang Ah Lee, Tae-Won Kwon, Yong-Pil Cho.

**Investigation:** Eunae Byun, Pil Je Kang, Sung Ho Jung, Seo Young Park, Sang Ah Lee, Tae-Won Kwon, Yong-Pil Cho.

**Methodology:** Eunae Byun, Pil Je Kang, Sung Ho Jung, Seo Young Park, Sang Ah Lee, Yong-Pil Cho.

**Resources:** Yong-Pil Cho.

**Supervision:** Pil Je Kang, Sung Ho Jung, Tae-Won Kwon, Yong-Pil Cho.

**Validation:** Eunae Byun, Yong-Pil Cho.

**Writing – original draft:** Eunae Byun, Seo Young Park, Sang Ah Lee, Yong-Pil Cho.

**Writing – review & editing:** Yong-Pil Cho.

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
