## [Decision Letter · Decision Letter 0]

20 Dec 2023

PONE-D-23-28460Impact of extracorporeal membrane oxygenation-related complications on in-hospital mortalityPLOS ONE

Dear Dr. Cho,

Thank you for submitting your manuscript to PLOS ONE. After careful consideration, we feel that it has merit but does not fully meet PLOS ONE’s publication criteria as it currently stands. Therefore, we invite you to submit a revised version of the manuscript that addresses the points raised during the review process.

We look forward to receiving your revised manuscript.

Kind regards,

Chiara Lazzeri

Academic Editor

PLOS ONE

Journal Requirements:

  a) The name of the colleague or the details of the professional service that edited your manuscript.

  b) A copy of your manuscript showing your changes by either highlighting them or using track changes (uploaded as a *supporting information* file).

  c) A clean copy of the edited manuscript (uploaded as the new *manuscript* file).

3. We note that there is identifying data in the Supporting Information file <ECMO raw data.xlsx>. Due to the inclusion of these potentially identifying data, we have removed this file from your file inventory. Prior to sharing human research participant data, authors should consult with an ethics committee to ensure data are shared in accordance with participant consent and all applicable local laws.

-Location data

Reviewers' comments:

Reviewer's Responses to Questions

**Comments to the Author**

1. Is the manuscript technically sound, and do the data support the conclusions?

Reviewer #1: Yes

Reviewer #2: Yes

2. Has the statistical analysis been performed appropriately and rigorously? 

Reviewer #1: Yes

Reviewer #2: Yes

3. Have the authors made all data underlying the findings in their manuscript fully available?

Reviewer #1: Yes

Reviewer #2: Yes

4. Is the manuscript presented in an intelligible fashion and written in standard English?

Reviewer #1: Yes

Reviewer #2: Yes

5. Review Comments to the Author

Reviewer #1: This is a retrospective study presented by Byun et al on the impact of ECMO related complications on in hospital mortality. It specifically focuses on vascular complications and cerebrovascular complications and compares the incidences between VA-ECMO and VV-ECMO. The overall concept of the study is well thought and the analyses are applied well. Within the realm of critical care and mechanical circulatory support this study is very relevant. Below are my feedback points for consideration:

- The title does not reflect what the study about; the study focused on comparing vascular and cerebrovascular complication in VA- versus VV-ECMO. The title should be updated to better reflect this.

- The study points out that the morbidity and mortality among ECMO patients is high but it doesn't outline that all ECMO patients are critically ill by the time they require this intervention whether it is for VA- or VV-ECMO. This point should be clarified.

- Lines 95 and 97 where the dates of study are included are confusing and should be rephrased to better reflect that the patients included in the study were over a 4 year span from 2015-2019.

- The study included patients who had undergone both VA- and VV-EMCO. Since this study heavily compares outcomes between VA and VV ECMO it would be more appropriate to exclude these incidences because it is not possible to parse whether the outcomes observed are related to one or the other even if the ECMO events were isolated in time.

- thromboembolic events were defined by medical diagnosis of limb ischemia, intracardiac or aortic thrombosis but all of these diagnoses are typically confirmed by imaging (CT, ultrasound), why was this not used

- For the cerebrovascular events the study says that an expert neurologist confirmed embolic or hemorrhagic strokes, were these diagnoses made by a clinical neurologist at the time of hospitalization or post-hoc by a study neurologist? Could there have been potential for bias here?

- ECMO can be deployed as a bridge to definitive treatment, such as transplantation or VAD placement, or as a replacement therapy for cardiac or respiratory failure; was this difference in indication accounted for looking at differences in outcomes? Bridging patients tend to be less severely ill than acute end organ failure patient and could represent a confounding factor within these analyses.

- Line 212, it's highlighted that a history of CAD had a protected effect on the occurrence of vascular complications; there is no direct relationship in the pathophysiology between the two and caution against calling this statistical observation a clinical protective effect. Secondary explanations should be explored, such as were CAD patients on medical therapy that were more likely to protect the patient from vascular complications?

- Were demographic factors such as history of prior surgery or types of medications that patients were on taken into consideration with the analyses in this study?

- A discussion about how VA versus VV cannulation differs is needed to better understand how certain complications may be more associated with one arrangement than another; additionally, the study would benefit from including a figure diagramming circuit configuration of VA and VV ECMO since this is not something readily known by most readers without a cardiac surgery background.

Reviewer #2: The authors present a single center retrospective cohort study to investigate the association of ECMO-related complications and in-hospital mortality, specifically vascular and cerebrovascular complications. The authors should be congratulated for their work, as they present a high volume of cases with good study design and surprising results. Overall the manuscript is well written. In their work, they find that vascular and cerebrovascular complications while on ECMO are not independent risk factors for in-hospital mortality.

The overall event rate between vascular and cerebrovascular events were high (20.2% and 13.6%). Vascular events were higher in VA ECMO, and had a trend toward higher incidence with cerebrovascular events with VA ECMO. Overall it was surprising that, in particular, cerebrovascular events were not associated with in-hospital mortality.

Major

- Overall the data is interesting, however there are some aspects of the various regression models that seem counterintuitive, and I wonder if it is more an issue with the models rather than what is truly happening with the data. For example, increased age and the presence of CAD were protective against vascular events. Some of the confidence intervals were extremely narrow (for example, ECMO running time), or were 0 (for example CPCR in table S6 with OR 0.00 and CI 0.00-0.00. This could perhaps be better explained in the discussion.

- It would be interesting to know the breakdown of ischemic versus hemorrhagic strokes. In our experience, hemorrhagic strokes portend worse outcomes.

- I appreciate the authors acknowledging the lack of data on functional outcomes in their cohort in the discussion. It would be beneficial to know if those who survive with cerebrovascular events are discharged with poor functional outcomes to have a better understanding of the clinical implication of these events.

Minor

Abstract:

- Line 31: Would use the term ECMO runs, or patients supported with ECMO, rather than ECMOs. Would use this terminology throughout the manuscript.

- Line 38: VA ECMO instead of ECMOs. VV ECMO instead of ECMOs.

- Line 39: The term cardiopulmonary cerebral resuscitation may be confusing to readers. I would change this to either cardiopulmonary resuscitation, or if patients were not ventilated, then cardio-cerebral resuscitation. This will require additional changes throughout the manuscript.

- Line 43: Again, would change ECMOs to ECMO.

Introduction and Methods: These sections are well written.

Results:

- In the univariate analysis, age with OR of 0.99 and very narrow confidence interval seems odd. This would suggest older age is protective against vascular complications in ECMO. (table 2)

- The odds ratio and confidence intervals for PAOD and CPCR in table S6 are 0.00. This does not seem correct.

Discussion:

- Line 380-381: Cesar trial randomized to referral to an ECMO center. 24% never received ECMO. I am not sure with Cesar trial it can be said that ECMO was superior. I believe this was better demonstrated with EOLIA and subsequent post-hoc analyses.

6. PLOS authors have the option to publish the peer review history of their article (what does this mean?). If published, this will include your full peer review and any attached files.

Reviewer #1: No

Reviewer #2: No

---

## [Author Response · Author response to Decision Letter 0]

1 Mar 2024

Ref. No.: PONE-D-23-28460

Title: Impact of extracorporeal membrane oxygenation-related complications on in-hospital mortality

PLOS ONE

Dear Academic Editor:

We appreciate your valuable comments regarding our manuscript entitled, "Impact of extracorporeal membrane oxygenation-related complications on in-hospital mortality," and the opportunity to revise it for potential publication in PLOS ONE.

We have addressed your concerns and comments and have included (below) our detailed responses to each specific point raised by the editor and reviewers. We have taken into account the editor's and reviewers' suggestions and have uploaded two revised versions of the manuscript—one with markup showing all of the changes made to our original version, and the other, the unmarked version of our revised manuscript.

We appreciate all of the comments and look forward to working with you toward the publication of our work in PLOS ONE.

Thank you for your kind consideration, and we look forward to hearing from you again.

Sincerely,

Yong-Pil Cho, MD, PhD 

Response to Editor's comments:

Reply: We included the aforementioned items when submitting our revised manuscript. Thank you.

Response to Journal Requirements:

 a) The name of the colleague or the details of the professional service that edited your manuscript.

 b) A copy of your manuscript showing your changes by either highlighting them or using track changes (uploaded as a *supporting information* file).

 c) A clean copy of the edited manuscript (uploaded as the new *manuscript* file).

3. We note that there is identifying data in the Supporting Information file <ECMO raw data.xlsx>. Due to the inclusion of these potentially identifying data, we have removed this file from your file inventory. Prior to sharing human research participant data, authors should consult with an ethics committee to ensure data are shared in accordance with participant consent and all applicable local laws.

-Location data

-ID numbers that seem specific (long numbers, include initials, titled "Hospital ID") rather than random (small numbers in numerical order)

Reply: We have addressed the journal's requirements. We have received a professional scientific editing service and uploaded the certificate as a *supporting information* file. We also re-uploaded a fully anonymized data set. Thank you.

Comments to the Author

1. Is the manuscript technically sound, and do the data support the conclusions?

Reviewer #1: Yes

Reviewer #2: Yes

2. Has the statistical analysis been performed appropriately and rigorously?

Reviewer #1: Yes

Reviewer #2: Yes

3. Have the authors made all data underlying the findings in their manuscript fully available?

Reviewer #1: Yes

Reviewer #2: Yes

4. Is the manuscript presented in an intelligible fashion and written in standard English?

Reviewer #1: Yes

Reviewer #2: Yes

Response to Reviewers' comments:

Reviewer #1: This is a retrospective study presented by Byun et al on the impact of ECMO related complications on in hospital mortality. It specifically focuses on vascular complications and cerebrovascular complications and compares the incidences between VA-ECMO and VV-ECMO. The overall concept of the study is well thought and the analyses are applied well. Within the realm of critical care and mechanical circulatory support this study is very relevant. Below are my feedback points for consideration:

Thank you for your kind words; we do our best to reflect our findings with careful analysis.

- The title does not reflect what the study about; the study focused on comparing vascular and cerebrovascular complication in VA- versus VV-ECMO. The title should be updated to better reflect this.

Reply: Thank you for pointing this out. However, we find the title describes our findings adequately; our purpose was to explore and share our results after treating each case as separate data. In this manuscript, we have concluded that "ECMO-related vascular and cerebrovascular complications were not associated with an increased risk of in-hospital death among adult patients." We hope this addresses your concerns.

- The study points out that the morbidity and mortality among ECMO patients is high but it doesn't outline that all ECMO patients are critically ill by the time they require this intervention whether it is for VA- or VV-ECMO. This point should be clarified.

Reply: Thank you for your comments. We have added some descriptions to the Introduction section. We hope this answer suffices.

- Lines 95 and 97 where the dates of study are included are confusing and should be rephrased to better reflect that the patients included in the study were over a 4 year span from 2015-2019.

Reply: Thank you for your comment. The lines simply state that the medical data of patients who have received ECMO for 5 years were collected between May 18, 2020, and May 17, 2021. We recognize that this can be confusing; we have deleted this sentence.

- The study included patients who had undergone both VA- and VV-EMCO. Since this study heavily compares outcomes between VA and VV ECMO it would be more appropriate to exclude these incidences because it is not possible to parse whether the outcomes observed are related to one or the other even if the ECMO events were isolated in time.

Reply: Thank you for your comment. The study seeks to observe the impact of each medical procedure (i.e., VA and VV ECMO) and share the findings that were present at the end of each ECMO care. Pre-procedural baseline data and their respective post-procedural data were collected for every ECMO case. As indicated in the 'Study design and study sample' of the Materials and Methods section, our analysis is based on a per-ECMO. Each ECMO procedure was analyzed as independent data. We hope this answer suffices.

- thromboembolic events were defined by medical diagnosis of limb ischemia, intracardiac or aortic thrombosis but all of these diagnoses are typically confirmed by imaging (CT, ultrasound), why was this not used

Reply: We agree with your points. In this study, thromboembolic events were verified with ultrasonography or computed tomographic (CT) scans. We have added some descriptions to 'the Definitions and study outcomes' of the Materials and Methods section, where we describe definitions of the study outcomes.

- For the cerebrovascular events the study says that an expert neurologist confirmed embolic or hemorrhagic strokes, were these diagnoses made by a clinical neurologist at the time of hospitalization or post-hoc by a study neurologist? Could there have been potential for bias here?

Reply: Despite this being a retrospective study subject to selection bias, all patients are subject to our clinical neurologists' diagnoses during their hospitalization.

- ECMO can be deployed as a bridge to definitive treatment, such as transplantation or VAD placement, or as a replacement therapy for cardiac or respiratory failure; was this difference in indication accounted for looking at differences in outcomes? Bridging patients tend to be less severely ill than acute end organ failure patient and could represent a confounding factor within these analyses.

Reply: Thank you for your comments. We agree with your points. However, as described in the Discussion section, this was a single-center study with a relatively small sample size, precluding detailed subgroup analyses, and thus, it is likely to be underpowered for some outcomes. Furthermore, because of various indications for ECMO support, we could not analyze the relationship between indications for ECMO support and outcomes. In this study, we did not look into the results where ECMO was used as a bridge to definitive treatment.

- Line 212, it's highlighted that a history of CAD had a protected effect on the occurrence of vascular complications; there is no direct relationship in the pathophysiology between the two and caution against calling this statistical observation a clinical protective effect. Secondary explanations should be explored, such as were CAD patients on medical therapy that were more likely to protect the patient from vascular complications?

Reply: I am sorry, but we made a mistake in statistical analysis (Table 2). A P value of a history of CAD from univariable analysis is 0.81, and, therefore, we did not include a history of CAD in the multivariable model (line 197, Table 2). We revised our manuscript, accordingly.

- Were demographic factors such as history of prior surgery or types of medications that patients were on taken into consideration with the analyses in this study?

Reply: Given the different indications for ECMO support and the diversity of each patient's underlying disease, we could not analyze the history of prior surgery or types of medications in this study.

- A discussion about how VA versus VV cannulation differs is needed to better understand how certain complications may be more associated with one arrangement than another; additionally, the study would benefit from including a figure diagramming circuit configuration of VA and VV ECMO since this is not something readily known by most readers without a cardiac surgery background.

Reply: We added schematic representative figures of VA and VV ECMO configurations as S1 Fig.

We appreciate your kind consideration.

Reviewer #2: The authors present a single center retrospective cohort study to investigate the association of ECMO-related complications and in-hospital mortality, specifically vascular and cerebrovascular complications. The authors should be congratulated for their work, as they present a high volume of cases with good study design and surprising results. Overall the manuscript is well written. In their work, they find that vascular and cerebrovascular complications while on ECMO are not independent risk factors for in-hospital mortality.

The overall event rate between vascular and cerebrovascular events were high (20.2% and 13.6%). Vascular events were higher in VA ECMO, and had a trend toward higher incidence with cerebrovascular events with VA ECMO. Overall it was surprising that, in particular, cerebrovascular events were not associated with in-hospital mortality.

Thank you for your kind words; we hope that this manuscript will offer some insights to those who are in similar fields.

Major

- Overall the data is interesting, however there are some aspects of the various regression models that seem counterintuitive, and I wonder if it is more an issue with the models rather than what is truly happening with the data. For example, increased age and the presence of CAD were protective against vascular events. Some of the confidence intervals were extremely narrow (for example, ECMO running time), or were 0 (for example CPCR in table S6 with OR 0.00 and CI 0.00-0.00. This could perhaps be better explained in the discussion.

Reply: Thank you for your comments. I am sorry, but we made a mistake in statistical analysis (Table 2). A P value of a history of CAD from univariable analysis is 0.81, and, therefore, we did not include a history of CAD in the multivariable model (line 197, Table 2). We revised our manuscript, accordingly. For the statistical analysis, we consulted a statistician (Ms. Seo Young Park, one of the coauthors of our manuscript). The explanation for OR 0.00 (CI 0.00�0.00, P < 0.00) is due to the small sample size. This translates to its 2 × 2 table, where the numbers represent its very small numbers. Regarding CPCR in S6 Table, there is one case of cerebrovascular event with CPCR, and no such event in eight cases. We hope this adequately addresses your question.

- It would be interesting to know the breakdown of ischemic versus hemorrhagic strokes. In our experience, hemorrhagic strokes portend worse outcomes.

Reply: Thank you for your comments. We agree with your points. We have repeated the statistical analysis as you recommended. There was no difference in the incidence of in-hospital death between ischemic versus hemorrhagic strokes (59/90, 65.6% versus 17/28, 60.7%, P = 0.66). Furthermore, in multivariable analysis, cerebrovascular complication is not a significant risk factor associated with in-hospital mortality (Table 3), and therefore, we did not revise our manuscript. We hope this addresses your point.

 Total VA ECMO VV ECMO P-value

Cerebrovascular event 116 (13.6) 103 (14.5) 13 (8.8) 0.09

ischemic 90 (10.5) 83 (11.7) 7 (4.8) 0.019

hemorrhagic 28 (3.3) 22 (3.1) 6 (4.1) 0.725

Categorical data are given as numbers (%)

 In-hospital death

ischemic 59/90 (65.6)

hemorrhagic 17/28 (60.7)

Values in parentheses are percentages.

- I appreciate the authors acknowledging the lack of data on functional outcomes in their cohort in the discussion. It would be beneficial to know if those who survive with cerebrovascular events are discharged with poor functional outcomes to have a better understanding of the clinical implication of these events.

Reply: Thank you for your comment.

Minor

Abstract:

- Line 31: Would use the term ECMO runs, or patients supported with ECMO, rather than ECMOs. Would use this terminology throughout the manuscript.

Reply: Thank you for pointing this out. We have revised our Abstract accordingly.

- Line 38: VA ECMO instead of ECMOs. VV ECMO instead of ECMOs.

Reply: We have revised our manuscript accordingly.

- Line 39: The term cardiopulmonary cerebral resuscitation may be confusing to readers. I would change this to either cardiopulmonary resuscitation, or if patients were not ventilated, then cardio-cerebral resuscitation. This will require additional changes throughout the manuscript.

Reply: In this study, all included patients were ventilated. Our working definition of cardiopulmonary cerebral resuscitation is a resuscitation carried out with medical equipment performed by health care professionals. Cardiopulmonary resuscitation, in our definition, can be enacted by anyone. We would like to keep our current definition, cardiopulmonary cerebral resuscitation. We hope this addresses your concern.

- Line 43: Again, would change ECMOs to ECMO.

Reply: We have revised our manuscript accordingly.

Introduction and Methods: These sections are well written.

Results:

- In the univariate analysis, age with OR of 0.99 and very narrow confidence interval seems odd. This would suggest older age is protective against vascular complications in ECMO. (table 2)

Reply: For the statistical analysis, we consulted a statistician (Ms. Seo Young Park, one of the coauthors of our manuscript). She repeated the statistical analysis; in multivariable analysis, older age is not protective against vascular complications in ECMO. We hope this answer sufficiently addresses your concern.

- The odds ratio and confidence intervals for PAOD and CPCR in table S6 are 0.00. This does not seem correct.

Reply: Thank you for your comments. For the statistical analysis, we consulted a statistician (Ms. Seo Young Park, one of the coauthors of our manuscript). The explanation for OR 0.00 (CI 0.00�0.00, P < 0.00) is due to the small sample size. This translates to a 2 × 2 table where the numbers represent its very small numbers. We hope this addresses your concerns.

Discussion:

- Line 380-381: Cesar trial randomized to referral to an ECMO center. 24% never received ECMO. I am not sure with Cesar trial it can be said that ECMO was superior. I believe this was better demonstrated with EOLIA and subsequent post-hoc analyses.

Reply: Thank you for your comment. We added the EOLIA trial (Ref. no. 11) in the Reference section, as you recommended.

[11] Goligher EC, Tomlinson G, Hajage D, Wijeysundera DN, Fan E, Jüni P, et al. Extracorporeal membrane oxygenation for severe acute respiratory distress syndrome and posterior probability of mortality benefit in a post hoc Bayesian analysis of a randomized clinical trial. JAMA. 2018;320:2251–2259. doi: 10.1001/jama.2018.14276.

Thank you for your comments. We appreciate your kind consideration.

Sincerely,

Yong-Pil Cho, MD, PhD

Division of Vascular Surgery, Department of Surgery, University of Ulsan College of Medicine and Asan Medical Center, Asanbyeongwon-gil 86, Songpa-gu, Seoul, Korea, 05505

Tel.: +82-2-3010-5039, Fax: +82-2-3010-6701, E-mail: ypcho@amc.seoul.kr

---

## [Editor Report · Decision Letter 1]

5 Mar 2024

Impact of extracorporeal membrane oxygenation-related complications on in-hospital mortality

PONE-D-23-28460R1

Dear Dr. Cho,

We’re pleased to inform you that your manuscript has been judged scientifically suitable for publication and will be formally accepted for publication once it meets all outstanding technical requirements.

Kind regards,

Chiara Lazzeri

Academic Editor

PLOS ONE
---

## [Editor Report · Acceptance letter]

15 Mar 2024

PONE-D-23-28460R1 

PLOS ONE

Dear Dr. Cho, 

I'm pleased to inform you that your manuscript has been deemed suitable for publication in PLOS ONE. Congratulations! Your manuscript is now being handed over to our production team.

Kind regards, 

on behalf of

Dr. Chiara Lazzeri 

Academic Editor

PLOS ONE